# Potential Roles of Iridoid Glycosides and Their Underlying Mechanisms against Diverse Cancer Growth and Metastasis: Do They Have an Inhibitory Effect on Cancer Progression?

**DOI:** 10.3390/nu13092974

**Published:** 2021-08-26

**Authors:** Cho-Won Kim, Kyung-Chul Choi

**Affiliations:** Laboratory of Biochemistry and Immunology, College of Veterinary Medicine, Chungbuk National University, Cheongju 28644, Chungbuk, Korea; rlachdnjs486@naver.com

**Keywords:** iridoids, phytochemicals, anticancer, invasion, angiogenesis, metastasis

## Abstract

Iridoids are glycosides found in plants, having inherent roles in defending them against infection by viruses and microorganisms, and in the rapid repair of damaged areas. The emerging roles of iridoid glycosides on pharmacological properties have aroused the curiosity of many researchers, and studies undertaken indicate that iridoid glycosides exert inhibitory effects in numerous cancers. This review focuses on the roles and the potential mechanism of iridoid glycosides at each stage of cancer development such as proliferation, epithelial mesenchymal transition (EMT), migration, invasion and angiogenesis. Overall, the reviewed literature indicates that iridoid glycosides inhibit cancer growth by inducing cell cycle arrest or by regulating apoptosis-related signaling pathways. In addition, iridoid glycosides suppress the expression and activity of matrix metalloproteinases (MMPs), resulting in reduced cancer cell migration and invasiveness. The antiangiogenic mechanism of iridoid glycosides was found to be closely related to the transcriptional regulation of pro-angiogenic factors, i.e., vascular endothelial growth factors (VEGFs) and cluster of differentiation 31 (CD31). Taken together, these results indicate the therapeutic potential of iridoid glycosides to alleviate or prevent rapid cancer progression and metastasis.

## 1. Introduction

### 1.1. Chemical Nature of Iridoid Glycosides

Recent years has seen a global increase in the consumption of plant-derived ingredients through functional foods, juices or pills, to aid in improving the general health. These chemical compounds produced from plants are called phytochemicals, and are known to exert beneficial effects such as antibacterial, anticancer, antioxidant, blood cholesterol lowering, immune function enhancement, and anti-aging [1,2,3,4]. Iridoids are glycosides found in various plants, and they reportedly bind to glucose [5]. They have the general form of cyclopentopyran, and a molecular structure related to iridodial (Figure 1A) [6,7]. Iridoids are structurally classified into iridoid glycosides and non-glycosidic iridoids according to the presence or absence of intramolecular glycosidic bonds; additionally, iridoid glycosides can be further subdivided into carbocyclic iridoids and secoiridoids [8]. The cleavage of the cyclopentane ring leads to secoiridoids, acting as a pharmacologically active ingredient similar to an iridoid [9]. The basic chemical structure of iridoids in plants (the iridoid ring scaffold) is biosynthesized in plants by the enzyme iridoid synthase using 8-oxogeranial as a substrate [8]. This plant-derived enzyme generates the iridoid ring scaffold through NADPH-dependent reduction and cyclization that occurs through Diels-Alder reaction or intramolecular Michael addition [8]. Figure 1 shows the chemical structure of iridodial and iridoid glycosides.

### 1.2. Biological Activities of Iridoid Glycosides

Iridoid glycosides produced in plants mainly defend against infections by viruses and microorganisms, and rapidly repair the damaged areas [10]. Additionally, iridoid glycosides are generally regarded as antifeedants against insects because of their deterrent bitter taste [11]. Surprisingly, iridoid glycosides isolated as active ingredients from traditional folk medicinal plants exhibit a wide range of pharmacological and physiological outcomes in the body [12]. Iridoid glycosides derived from various medicinal plants have shown therapeutic benefits in relation to diseases such as neurological disorders, diabetes mellitus and cardiovascular disorders, as well as cancers [10,13,14,15]. Recent studies suggest that iridoid glycosides may be considered as potential therapeutic targets for arthritis [16,17]. Additionally, one of the iridoid glycosides, aucubin, can protect the liver from poisoning caused by α-amanitin, and showed a strong preventive effect from CCl4-induced liver damage [18]. However, despite the iridoid glycosides are demonstrated to exert anticancer effects in numerous cancer types, the understanding of the role of iridoid glycosides at each stage of cancer development is still lacking.

The continuous accumulation of genetic mutations in normal cells results in cell mutations, and the consequent occurrence of cancer. A single cancer cell clone is capable of proliferating to form a tumor mass. In order to expand the growth area and receive continuous nutrients, the cancer cell acquires a highly mobile phenotype, forms blood vessels, and initiates a metastatic cascade [19,20]. Malignant tumor progression depends on the invasion, metastasis, and the ability to promote the host response to angiogenesis [21]. Understanding the mechanisms by which iridoid glycosides are capable of inhibiting each process of cancer development will reveal the target molecules of iridoid glycosides, thereby leading to the appropriate therapeutic use of these compounds.

In this review, we focus on the updated roles of iridoid glycosides at each stage of cancer development, such as proliferation, epithelial mesenchymal transition (EMT), migration, invasion and angiogenesis. This review also highlights the therapeutic potentials of iridoid glycosides for cancer.

## 2. Effects of Iridoid Glycosides on Cancer Development and Metastasis

### 2.1. Anti-Proliferative and Apoptotic Effects

In general, cells undergo strictly regulated processes of growth, differentiation, and programmed cell death (e.g., apoptosis or autophagy), or remain in a state of growth cessation [22]. However, abnormalities (chromosomal abnormalities, mutations, etc.) occurring in some genes of a cell result in altered properties of gene products, viz., proteins, and subsequent abnormalities in cell growth regulation [23]. These genetic mutations could accumulate for as little as several months or as long as 20–30 years, eventually mutating into cancer cells and proliferating to form tumors [24,25]. A major feature of cancer is that a single cells continues to proliferate abnormally and form tumors [20]. The continuous proliferation of cancer cells is the outcome of a deregulated cell cycle and inhibition of programmed cell death [26]. Abnormalities in signaling pathways that control cell proliferation and cell survival are essential for tumorigenesis, additionally, mutations of survival signaling pathways such as insulin-like growth factor (IGF) and Akt, or overexpression of anti-apoptotic oncoproteins such as the Bcl-2 family genes, increases cancer cell survival [27,28]. Therefore, manipulating the signal transduction molecules involved in the proliferation and death of cancer cells can help in the therapeutic improvement for cancer.

Catalpol is one of the main active ingredients of a traditional Chinese medicinal plant *Rehmannia glutinosa*, and exerts a pharmacological effect that mainly inhibits cancer growth and tumorigenesis [29]. This compound is one of the most studied iridoid glycosides due to its remarkable pharmacological effects. Gao et al. reported that catalpol significantly decreases the matrix metalloproteinase (MMP)-2 signaling, and increases the expression level of microRNA (miR)-200, which regulates proliferation, invasion and metastasis, thereby decreasing cell proliferation and accelerating apoptosis in the OVCAR-3 human ovarian cancer cell line [30]. Another study in breast cancer cell lines reported that catalpol downregulates MMP-16 expression and upregulates miR-146a expression, resulting in decreased proliferation of MCF-7 cells [29]. In addition, this compound has demonstrated anticancer efficacy by inducing apoptosis in the T24 bladder cancer cell line via the phosphoinositide 3-kinase (PI3K)/Akt pathway [31]. In two studies using an in vitro colorectal cancer model, catalpol was found to promote apoptosis and autophagy in colorectal cancer cells, either via the PI3K-Akt signaling pathway or by directly inhibiting sirtuin 1 (SIRT1) expression [32,33].

The main active ingredients of olive oil include phenolic constituents. Oleuropein, a major phenolic compound, is known to exhibit various pharmacological activities [13], and can be obtained from virgin olive oil before it is chemically removed, since it imparts a bitter taste to olive oil [34]. To date, several cancer cell lines studied previously revealed that oleuropein has anti-proliferative activity in blood cancer, lung cancer, cervical cancer, leukemia, and breast cancer. In the HL60 human promyelocytic cell line, virgin olive oil phenol at a concentration of 13.5 mg/L is reported to completely block cell proliferation through accumulation of cells in the G0/G1 phase, and induce apoptosis due to superoxide generation [35]. Oleuropein-induced anti-proliferative effects have also been observed in A549 human lung carcinoma cells, as observed by an increase in the number of cells entering the G1 phase of the cell cycle [36]. HeLa human cervical cancer cells were arrested at the G2/M phase by oleuropein treatment, and exposure to oleuropein resulted in increased levels of phosphorylated ATF-2, c-Jun NH2-terminal kinase (JNK), p53, p21, Bcl-2-associated X protein (Bax) and cytochrome c protein, resulting in apoptosis [37]. Interestingly, hydroxytyrosol, one of the ester metabolites of oleuropein, showed significant inhibition of proliferation via the extracellular signal-regulated kinase (ERK)1/2-cyclin D1 pathway in MDA-MB-231 human breast adenocarcinoma [38]. The antitumor properties of hydroxytyrosol were also demonstrated through G2/M cell cycle arrest in human hepatocellular carcinoma cells, and tumor growth inhibition in a hepatocellular carcinoma-inoculated orthotopic xenograft model [39]. Although hydroxytyrosol exerted no statistically significant effect on cell proliferation or apoptosis at the cellular level, it delayed growth of the HT-29 colorectal tumor xenograft in athymic nude mice [40].

Aucubin is one of the iridoid glycosides commonly found in plants and acts as a protective compound, as determined by the anti-proliferative effects in two types of cancer models [41]. Aucubin exhibited anti-leukemic activity in K562 cells, and hydrolyzed aucubin also inhibited proliferation of K562 human chronic myeloid leukemia (CML) cells through cell cycle regulation by inhibiting cells in the sub-G1 phase [42,43]. The hydrolyzed form inhibits the BCR–ABL phosphorylation and induces apoptosis in CML cells; surprisingly, the authors concluded that hydrolyzed aucubin had a better anti-leukemia effect than aucubin itself [43]. CML is a myeloproliferative disease mainly caused by BCR–ABL gene fusion. Suppressing the expression of BCR–ABL protein suggests the possibility that aucubin may be capable of targeting the underlying cause of CML [43,44]. In addition, aucubin treatment induced the A549 human non-small cell lung cancer (NSCLC) cell line to enter the G0/G1 phase, arresting cell cycle progression and inducing apoptosis through p53 and Fas and Fas ligand (FasL) signaling, thereby suggesting its involvement in the anti-proliferative activity of lung cancer [45].

Saracoglu et al. evaluated the anticancer effects of various iridoid glycosides isolated from *Veronicas* (*Speedwells*) in Hep-2 human epidermoid carcinoma, RD human rhabdomyosarcoma, L-20B transgenic murine L-cells, and Vero African green monkey kidney cells [46]. They argued that each iridoid glycoside exhibits cytostatic and apoptotic activity, depending on the chemical structure and type of cancer cell [46]. Another research team analyzed the molecular structures of seven iridoid glycosides obtained from the root of *Phlomoides umbrosa* Kamelin & Makhm using nuclear magnetic resonance (NMR), and the effect of these iridoid glycosides was examined on the cell viability of HeLa human cervical cancer cells, HL-60 human promyelocytic leukemia cells, and MCF-7 breast cancer cells [47]. Results confirmed that phlomisu E had the strongest cytotoxicity against all three cancer cell types, suggesting that there is a structure-activity relationship between iridoid glycosides and cytotoxicity [47].

The antitumor properties of an iridoid glycoside were also confirmed in animal models transplanted with breast cancer cells. The mechanism of anti-proliferative effects on MCF-7 and triple-negative breast cancer (TNBC) cell lines (MDA-MB-231, MDA-MB-453 and MDA-MB-468) by Jatamanvaltrate was elucidated at the cellular level by downregulation of cell cycle-related genes, apoptosis induced by enhancement of cleavage of PARP, and autophagy induced by increased LC3-II levels [48]. Consistent with these in vitro experimental data, Jatamanvaltrate further demonstrated antitumor activity due to apoptosis and autophagy in a subcutaneously inoculated xenograft mouse model of MDA-MB-231 breast cancer cells [48].

These results indicate that iridoid glycosides may inhibit cancer proliferation by upregulating the cell cycle arrest genes such as p53 and p21 or by causing accumulation of cells in the G0/G1 phase. The mechanism by which iridoid glycosides inhibit cancer cell proliferation also involves downregulation of the PI3K/Akt pathway and the ERK1/2-cyclin D1 pathway, and the upregulation of Bax and cytochrome c by iridoid glycosides indicates the possibility of programmed cell death due to apoptosis.

### 2.2. Inhibitory Effects on Epithelial-Mesenchymal Transition

Cancer cells have unregulated cell proliferation in their early stages, but an evolutionarily conserved developmental program (EMT) is associated with metastasis and peculiarizes metastatic properties in cancer cells by enhancing the cell mobility, invasiveness and resistance to programmed cell death [49,50,51]. EMT is known to be the main factor during the early stage of dissemination in most cancer types [52,53]. Through this morphological or epigenetic modification process, epithelial cells (which are strongly bound between cells) are converted to mesenchymal cells (which migrate easily), allowing cancer cells to migrate and penetrate other tissues [2,54,55]. Reduced expression of cell adhesion molecules such as epithelial cadherin (E-cadherin) allows cancer cells to act independently of other cells and tissue components, making it easier for cells to invade and metastasize [20]. When migrating through the bloodstream or lymphatic vessels to reach other tissues, the cancer cells undergo mesenchymal-epithelial transition (MET), a reverse process of EMT, wherein they are converted into epithelial cells that strongly bond with the surrounding cells, creating a tumor microenvironment favorable for colonization [50,51,56]. Thus, EMT and MET are reversible processes and can occur repeatedly in any sequence during the progress of metastasis.

Catalpol exerted an inhibitory effect on EMT in lung cancer, hepatocellular cancer and osteosarcoma cell lines [57,58]. Transforming growth factor (TGF)-β is known to play a critical role in overall tumor progression, including EMT, and due to the strong anti-tumoral effect of TGF-β inhibitor, anticancer drugs containing this inhibitor as the main component are widely applied clinically [59,60,61]. Catalpol suppresses TGF-β1-stimulated EMT in A549 human NSCLC cells through inactivation of the Smad2/3 and nuclear factor kappa-light-chain-enhancer of activated B cells (NF-κB) signaling pathways [57]. miR-140-5p is known to regulate the cell proliferation and migration ability in several carcinomas, and was observed to be reduced after TGF-β1 exposure in Huh7 and HCCLM3 hepatocellular carcinoma cell lines. However, treatment with catalpol reversed these TGF-β1 effects, upregulated the epithelial marker E-cadherin, and downregulated the expressions of mesenchymal markers vimentin and N-cadherin [62,63,64]. Another study with osteosarcoma cells revealed the mechanism by which catalpol inhibits cell proliferation and EMT, by targeting various molecules involved in cancer progression [58]. Catalpol inhibited EMT progression through downregulation of the rho associated coiled-coil containing protein kinase 1 (ROCK1) and MMP-2 expression in MG63 and U2OS human osteosarcoma cancer cell lines, and significantly reduced tumor growth in a dose-dependent manner in a xenograft model transplanted with MG63 cells [58].

It was suggested that oleuropein inhibits the EMT process in breast cancer cells by inducing upregulation of the epithelial marker E-cadherin, and downregulation of mesenchymal markers MMP-2 and MMP-9. The same study also confirmed that oleuropein significantly reduces the expression of an EMT-inducer transcription factor zinc finger E-box binding homeobox 1 (ZEB1) in breast cancer cells [53].

The iridoid glycosides such as catalpol and oleuropein have been shown to induce changes in these diverse genes related to EMT, which may promote morphological changes that make cancer cells more migratory. However, since the effect of iridoid glycosides on EMT has been identified in a small number of iridoid glycosides, it needs to be studied in more types of glycosides.

### 2.3. Inhibitory Effects on Cancer Migration and Invasion

Cancer cells that have undergone EMT are accompanied by cytoskeletal changes, and increased cell individualization and mobility. [65,66]. These cells promote cell elongation and motility by reorganizing the actin cytoskeleton [67,68]. Lamellipodia present at the leading edge of the cell and undergo a repetitive contraction-relaxation cycle with the help of filopodia, allowing the cell to translocate [69,70]. In order for cells to become invasive, subsequent genetic and morphological modifications are required. Cancer cells promote degradation of the extracellular matrix (ECM) by expressing genes related to MMPs and proteolytic activity, and the dynamic actin-rich invadopodia facilitate cell invasion by degrading the surrounding ECM [71,72,73]. Through these processes, cancer cells enter the circulatory system, including the bloodstream or lymphatic vessels, and initiate the metastatic cascades [21].

In several studies, exposure to catalpol reduced the mobility and migration by regulating miR expression in hepatocellular carcinoma. Catalpol inhibited the invasion of hepatocellular carcinoma cells by regulating the miR-22-3p/MTA3 axis; moreover, the expression of miR-140-5p was also found to be associated with inhibition of the invasion and migration of hepatocellular carcinoma cells [62,74]. It was further confirmed that catalpol inhibits the migration of MKN-45 human gastric cancer cells by inhibiting the expressions of MMP-2, α-smooth muscle actin (α-SMA), and ras homolog gene family member A (RhoA)-ROCK1 signaling pathways [75]. Furthermore, catalpol showed potential for anti-invasion by inhibiting the expressions of MMP-2 and MMP-9 in CT26 murine colorectal carcinoma cells [76].

Other studies have reported the anti-migration activity of oleuropein in different cancers. In a study on the incidence of skin cancer due to long-term UVB radiation, oleuropein inhibited the expressions of MMPs (MMP-2, MMP-9, MMP-13) involved in ECM remodeling and degrading, indicating the inherent potential to inhibit invasiveness [77]. Oleuropein also almost blocked the vertical and radial migration of cells in T-47D human breast cancer cells and RPMI-7951 human malignant melanoma cells [78].

Picroside I, Kutkoside, and Kutkin isolated from *Picrorhiza kurroa*, a traditional Chinese herb, showed anti-invasive activity against MCF-7 breast cancer cells, and this inhibitory effect was attributed to downregulation of the activity of gelatinases (MMP-2 and MMP-9) and collagenases (MMP-1 and MMP-13) [79]. Valjatrate E isolated from *Valeriana jatamansi* Jones inhibited cancer migration and invasion by inactivating the mitogen-activated protein kinase (MAPK)/ERK signaling pathway, and decreasing the expression and secretion of MMPs in human hepatocellular carcinoma HepG2 [80].

These results suggest that iridoid glycosides mainly suppress the expression and activity of MMPs, thereby lowering the proteolytic activity for ECM, resulting in significantly reduced cancer mobility and invasiveness. The blockade of the MAPK/ERK signaling pathway might increase this migration.

### 2.4. Anti-Antiangiogenic Effects

Angiogenesis plays an essential role in tumor development and growth, and is required for invasive tumor growth and metastasis [81]. The purpose and mechanisms of angiogenesis in cancer development and metastasis can be explained by two methods. First, the formation of blood vessels is essential in order for the tumor to continuously receive nutrients in the host body [81,82]. As the tumor grows, the density of cells at the center of the tumor increases in proportion to the size of the tumor, but the growth is limited by receiving nutrients and exchanging gases only through diffusion around the cells without angiogenesis [83,84,85]. In other words, avascular tumors may regress due to lack of adequate blood supply. Secondly, tumors promote angiogenesis to increase the likelihood of metastasis [85]. Angiogenic factors such as vascular endothelial growth factors (VEGFs) produced in tumor cells stimulate the secretion of enzymes that degrade the basement membrane by binding to receptors on the surface of surrounding endothelial cells [21,86]. Subsequently, tiny pores are formed between endothelial cells that form blood vessels, and through these pores, the endothelial cells grow towards the tumor, eventually forming a new vessel that connects the tumor and the blood vessels of the host [21]. This mechanism induces the production of MMPs and facilitates their migration through the ECM [87,88]. Lymphangiogenesis by VEGFs (e.g., VEGF-C) is induced by a mechanism similar to angiogenesis, and the lymphatic system can be a second route for tumor metastasis [89,90]. Therefore, inhibiting angiogenesis and shrinking the existing tumor blood vessels to delay tumor development and minimize metastasis, can be a promising anticancer strategy [81,91].

Exposure to catalpol represses tube formation in the human umbilical vein endothelial cells (HUVECs) cultured in CT26 supernatants, and inhibits aortic ring angiogenesis in rats, thereby indicating that catalpol exerts anti-angiogenetic properties against colon cancer [76]. In addition, catalpol inhibits the migration and tube formation of HUVECs and suppresses corneal neovascularization in rats; this mechanism includes regulation in the expressions of VEGF and an endogenous anti-angiogenic factor, viz., the pigment epithelium-derived factor (PEDF) [92].

The effects of oleuropein on angiogenesis has been studied in numerous cancer types. In chronic UVB-induced skin cancer tissues, administration of oleuropein not only significantly inhibited the diameter of subcutaneous blood vessels in ultraviolet B (UVB)-irradiated mice, but also lowered the expressions of VEGF, cluster of differentiation 31 (CD31), and cyclooxygenase-2 (COX-2), which play pivotal roles in angiogenesis [77]. Hydroxytyrosol, a metabolite of oleuropein, also exhibited anti-angiogenic activities in colorectal cancer and hepatic cancer. Hydroxytyrosol induced functional impairment of the hypoxia inducible factor-1alpha (HIF-1α)/microsomal prostaglandin-E synthase-1 (mPGEs-1)/PGE-2/VEGF axis in HT-29 and WiDr human colorectal adenocarcinoma cells [40]. Additional to the downregulation of VEGF and mPGEs-1 in vivo, the morphology of blood vessels was also modified with reduced blood perfusion to the tumor, indicating that hydroxytyrosol downregulates VEGF, MAPK activation and PGE-2 [40]. Taken together, these results indicate the anti-angiogenic activity of the metabolite in colon cancer. Hydroxytyrosol also remarkably downregulated the expression of CD31 (a pro-angiogenic factor) in HepG2- or Huh7-transplanted orthotopic hepatocellular carcinoma [39].

Picroside II, one of the major pharmacological components in *Picrorhiza kurroa*, is also reported to suppress tube formation of HUVECs, leading to noticeable inhibition of angiogenesis in the chorioallantoic membrane of chick embryos [93]. Similarly, β-hydroxyipolamiide, ipolamiide, and buddlejoside A5 isolated from *Stachys ocymastrum* and *Premna resinosa* were also confirmed to have anti-angiogenic effects, as demonstrated in zebrafish embryos and chick chorioallantoic membrane assays, supporting the evidence for the potential of iridoid glycoside to inhibit tumor angiogenesis [94].

Overall, these in vitro and in vivo studies have demonstrated that iridoid glycosides inhibit angiogenesis by down-regulating the expression of pro-angiogenic factors such as VEGF and CD31. Thus, iridoid glycosides inhibit tumor angiogenesis, impeding the continued growth of cancer cells, and may block the pathways that allow metastasis to distant organs. Table 1 summarizes the inhibitory effects and mechanisms of iridoid glycosides at each stage of cancer development.

## 3. Conclusions and Future Perspectives

This review provides a comprehensive understanding of studies undertaken on the anticancer effects of different iridoid glycosides present in herbal medicines or functional foods, and indicates the therapeutic potential for various cancers by understanding the functional roles and regulatory mechanisms of iridoid glycosides at each stage of cancer development. Overall, the reviewed literature indicates that iridoid glycosides inhibit cancer proliferation by inducing cell cycle arrest or down-regulating the PI3K/Akt pathway and the ERK1/2-cyclin D1 pathway, leading to programmed cell death or cytotoxicity. In addition, iridoid glycosides have been shown to induce changes in diverse genes related to EMT, and they suppress the expression and activity of MMPs, thereby lowering the proteolytic activity for ECM and resulting in significantly reduced cancer mobility and invasiveness. Moreover, the antiangiogenic mechanism of iridoid glycosides is closely related to the transcriptional regulation of pro-angiogenic factors such as VEGF and CD31. When considered together, these results indicate the therapeutic potential of iridoid glycosides to alleviate or prevent cascades of cancer development and metastasis.

In addition to iridoid glycosides, iridoid derivatives such as 8-acetylharpagide and genipin are reported to exhibit anti-proliferative and anti-metastatic effects on cancer [95,96]. In particular, genipin is shown to be a potent inhibitor of the mitochondrial uncoupling protein 2 (UCP2), and a tumor suppressor in various cancers [95]. Derivatives have a chemical structure similar to that of the parent structure, but sometimes exhibit more potent pharmacological and biological activity than the parent compound [43,97,98,99]. Therefore, further studies are required to investigate the effects of iridoid derivatives on cancer.

As described in this review, decades of accumulated experimental data connote the possible therapeutic implications of iridoid glycosides for cancer. However, some questions remain unanswered:How do iridoid glycosides affect the tumor microenvironment?Can iridoid glycosides stimulate the immune system to suppress the development of cancer?Can iridoid glycosides inhibit lymphangiogenesis with respect to tumor metastasis?Can iridoid glycosides restrict growth and secondary metastasis through tumor dormancy?Can iridoid glycosides enhance the curative effect, acting as an adjuvant to existing anticancer drugs?

With the exception of one clinical approach (catalpol) [100], proof of the therapeutic effect of most iridoid glycosides is limited to in vitro and in vivo studies. Based on reasonable evidence, research on the possible effective dose for cancer patients will enable the active use for implementing the chemopreventive and chemotherapeutic effects of iridoid glycosides.

## Figures and Tables

**Figure 1 nutrients-13-02974-f001:**
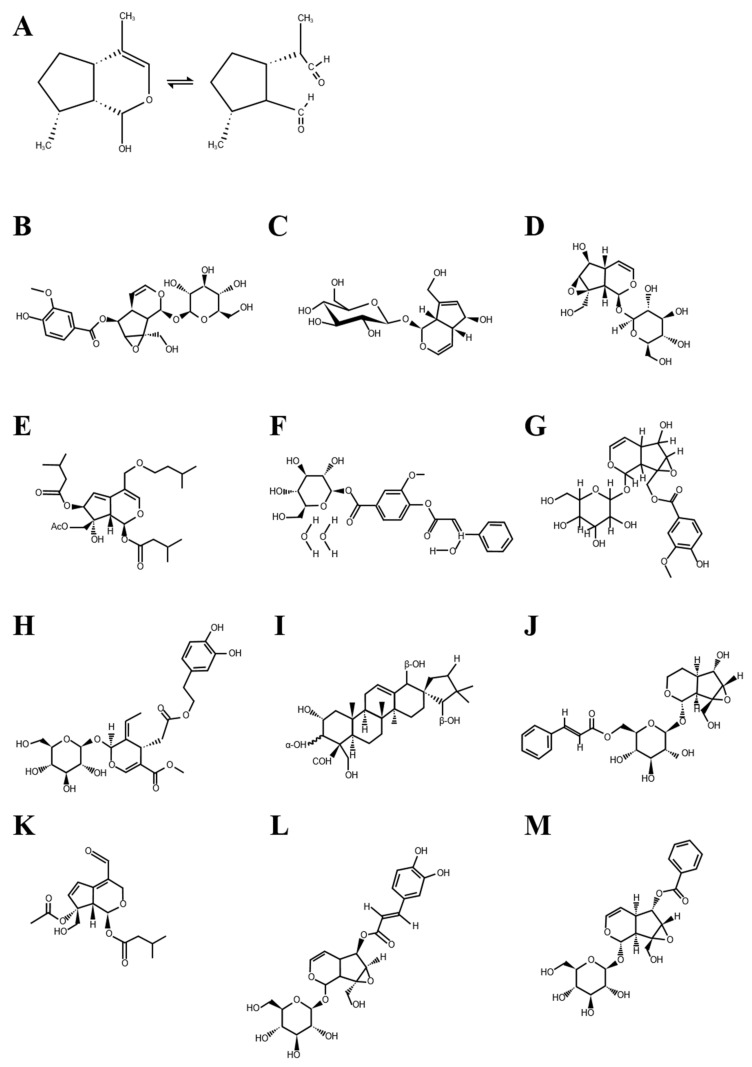
Chemical structure of iridodial and iridoid glycosides. (**A**) Iridodial, (**B**) Amphicoside, (**C**) Aucubin, (**D**) Catalpol, (**E**) Jatamanvaltrate P, (**F**) Kutkin, (**G**) Kutkoside, (**H**) Oleuropein, (**I**) Phlomisu, E (**J**) Picroside I, (**K**) Valjatrate E, (**L**) Verminoside and (**M**) Veronicoside.

**Table 1 nutrients-13-02974-t001:** Inhibitory effects and mechanisms of iridoid glycosides at each stage of cancer development.

Stages	Iridoid Glycosides	Effective Dosages	Key Effects and Inhibitory Mechanisms	Types of Cancer	In Vitro /In Vivo	Ref.
Proliferation	Catalpol	50 and 100 μg/mL	Reduces cell proliferation and accelerates apoptosis by regulating miR-200 and MMP-2 expressions	Ovarian cancer	In vitro	[30]
		50 and 100 μg/mL	Suppresses cell proliferation and facilitates apoptosis by regulating MMP-16 and miR-146a expressions	Breast cancer	In vitro	[29]
		80 and 160 μM	Suppresses cell proliferation and promotes apoptosis via the PI3K/Akt signaling pathway	Bladder cancer	In vitro	[31]
		50 and 100 μg/mL	Induces apoptosis through upregulation of miR-200, caspase-3 and -9, and downregulation of PI3K/Akt signaling	Colorectal cancer	In vitro	[33]
		30, 40 and 50 μM	Reduces cell proliferation and promotes apoptosis via the miR-34a/SIRT1 signaling pathway	Colorectal cancer	In vitro	[32]
	Oleuropein	12.5 and 25 μM	Suppresses cell proliferation through accumulation of cells in the G0/G1 phase Induces apoptosis by generating superoxide	Blood cancer	In vitro	[35]
		IC_50_ * = 59.96 μM	Inhibits cell proliferation by increasing the number of cells entering the G1 phase of the cell cycle	Lung cancer	In vitro	[36]
		150 and 200 μM	Inhibits cell proliferation by arresting the G2/M phase Induces apoptosis by increasing pATF-2, JNK, p53, p21, Bax and cytochrome c expressions, and activating caspase-3 and -9	Cervical cancer	In vitro	[37]
	Oleuropein (hydroxytyrosol **)	100 and 200 μM	Suppresses cell proliferation via the ERK1/2-cyclin D1 signaling pathway	Breast cancer	In vitro	[38]
		100, 200, 300 and 400 μM 10 and 20 mg/kg bw (i.p.)	Inhibits cell proliferation in vitro and tumor growth in vivo	Hepatocellular carcinoma	Both	[39]
		10 mg/kg bw (i.p.)	Delays the growth of HT-29 colorectal tumor xenograft in athymic nude mice	Colorectal cancer	In vivo	[40]
	Aucubin	IC_50_ = 44.7 μM	Inhibits cell proliferation	Chronic myelogenous leukemia	In vitro	[42]
		100, 150 and 200 μM	Suppresses cell proliferation through accumulation of cells in the sub-G1 phase, and downregulation of BCR–ABL and STAT3 Induces apoptosis by activating caspase-3 and by suppressing JAK2 and c-Src activation	Chronic myelogenous leukemia	In vitro	[43]
		1, 5, 10 and 20 μM	Blocks proliferation by upregulating the expressions of p53 and p21 Facilitates apoptosis by inducing the activation of the Fas/FasL signaling	Non-small cell lung cancer	In vitro	[45]
	Amphicoside	IC_50_ = 340 μM (Epidermoid carcinoma)	Increases cytotoxic and cytostatic activity	Epidermoid carcinoma Rhabdomyosarcoma	In vitro	[46]
	Verminoside	IC_50_ = 128 μM (Epidermoid carcinoma) IC_50_ = 70 μM (Rhabdomyosarcoma)				
	Veronicoside	IC_50_ = 153.3 μM (Epidermoid carcinoma) IC_50_ = 355 μM (Rhabdomyosarcoma)				
	Phlomisu E	IC_50_ = 19.3 μM (Cervical cancer) IC_50_ = 8.4 μM (Leukemia) IC_50_ = 15.4 μM (Breast cancer)	Increases cytotoxic activity	Cervical cancer Leukemia Breast cancer	In vitro	[47]
	Jatamanvaltrate P	10, 20, 50 μM 15 mg/kg bw (i.p.)	Inhibits proliferation by inducing G2/M phase arrest Activates autophagy by triggering autophagosome formation and by increasing LC3-II levels Exhibits antitumor effect in xenografts	Breast cancer	Both	[48]
EMT	Catalpol	5 and 10 μM	Inhibits TGF-β1-induced EMT through the inactivation of Smad2/3 and NF-κB signaling pathways	Lung cancer	In vitro	[57]
		20, 40 and 80 μM	Inhibits EMT by downregulating RACK1 and MMP-2 expression	Osteosarcoma	In vitro	[58]
		50 μM	Inhibits EMT by regulating the expression of miR-140-5p	Hepatocellular carcinoma	In vitro	[62]
	Oleuropein	600 μg/mL	Suppresses EMT through downregulation of SIRT1	Breast cancer	In vitro	[53]
Migration/ Invasion	Catalpol	50 μM	Inhibits invasion by regulating the miR-22-3p/MTA3 axis	Hepatocellular carcinoma	In vitro	[74]
		50 μM	Inhibits migration and invasion by regulating the expression of miR-140-5p	Hepatocellular carcinoma	In vitro	[62]
		20, 40 and 80 μM	Inhibits migration by downregulating the expression of MMP-2, α-SMA, RhoA-ROCK1 signaling pathways	Gastric cancer	In vitro	[75]
		1.25, 2.5 and 5 μM	Suppresses invasion by inhibiting the expressions of MMP-2 and MMP-9	Colon cancer	In vitro	[76]
	Oleuropein	0.01 and 0.1%	Blocks vertical and radial migration	Breast cancer	In vitro	[78]
		25 mg/kg bw (p.o.)	Inhibits migration by downregulating the expressions of MMP-2, MMP-9, and MMP-13	Skin cancer	In vivo	[77]
	Picroside I	5 μM	Inhibits invasion by downregulating the activity of gelatinases (MMP-2 and MMP-9) and collagenases (MMP-1 and MMP-13)	Breast cancer	In vitro	[79]
	Kutkoside	5 μM				
	Kutkin	5 μM				
	Valjatrate E	3, 6 and 12 μg/mL	Inhibits migration and invasion by inactivating MAPK/ERK signaling pathway and by decreasing the expression and secretion of MMP-2 and MMP-9	Hepatocellular carcinoma	In vitro	[80]
Angiogenesis	Catalpol	1.25, 2.5 and 5 μM 7, 14, 28 mg/kg bw (p.o.)	Suppresses forming ability of HUVEC Exhibits anti-angiogenesis activity against colon cancer	Colon cancer	Both	[76]
	Oleuropein	25 mg/kg bw (p.o.)	Reduces the diameter of subcutaneous blood vessels in UVB-irradiated mice by downregulating the expression of MMPs (MMP-2, MMP-9, MMP-13), VEGF, CD31 and COX-2	Skin cancer	In vivo	[77]
	Oleuropein (hydroxytyrosol *)	50 and 100 μM 10 mg/kg bw	Reduces angiogenesis of endothelial cells via HIF-1α/mPGEs-1/PGE-2/VEGF axis Modifies the morphology of blood vessels and reduces blood perfusion to tumor by downregulating VEGF, MAPK and PGE-2 expressions	Colon cancer	Both	[40]
		10 and 20 mg/kg bw (i.p.)	Downregulates the expression of CD31 in transplanted orthotopic hepatocellular carcinoma	Hepatocellular carcinoma	In vivo	[39]

* The concentration of 50% cellular cytotoxicity of human tumor cells. ** Ester metabolites of oleuropein.

## Data Availability

Not applicable.

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
