# Peer review of "Potential Roles of Iridoid Glycosides and Their Underlying Mechanisms against Diverse Cancer Growth and Metastasis: Do They Have an Inhibitory Effect on Cancer Progression?"

_nutrients, 2021, doi:10.3390/nu13092974_

Round 1

Reviewer 1 Report

In this study, the authors focused on presenting iridoids as potential substances in the prevention and therapy of cancer. They mainly focused on the possibility of inhibiting the progression of various types of cancer, where references to in vitro studies were presented. Several important issues were discussed, incl. Anti-proliferative and apoptotic effects, Inhibitory effects on epithelial-mesenchymal transition, Inhibitory effects on cancer migration and invasion, Anti-antiangiogenic effects. The authors also highlighted many aspects that have not yet been addressed in teamt literature. Their exact list can be found in the conclusions. In my opinion, the work is interesting and contains key information on the mechanisms of anti-cancer action of natural substances. The article may be a good background for other researchers and a contribution to further research on not only iridoids but also other substances of natural origin. The work is well organized, however, in my opinion, the authors should pay more attention to the main subject of the work, which are iridoids. In my opinion, the introduction part should be divided into two sections. In the first part, the authors should devote more attention to the chemical nature of these compounds with the presentation of structural and derivative formulas and references to the species in which they occur. The second part should contain information about the biological activity of these compounds in connection not only with antitumor activity. 

Author Response

Aug 19, 2021

Editor-in Chief

Nutrients

Dear Editor-in Chief

The authors appreciate for valuable comments by the reviewers to complete the manuscript. The manuscript has been extensively revised based on the comments and a "Response to Reviewers" which addresses the individual comments of each reviewer has been attached.

Please find the revised manuscript (#nutrients-1329887.R1) entitled: “Potential roles of iridoid glycosides and their underlying mechanisms against diverse cancer growth and metastasis: Do they have an inhibitory effect on cancer progression?” by Kim CW and Choi KC, which we would like to re-submit as a review article for the Special Issue of the Nutrients.

 Special issue of “Antitumor Effects of Dietary Compounds”

Section of “Phytochemicals and Human Health”

I hope that everything will be found as required.

Yours Sincerely,

Kyung-Chul Choi, DVM, PhD

Laboratory of Veterinary Biochemistry and Immunology

College of Veterinary Medicine

Chungbuk National University

Cheongju, Chungbuk

361-763, Republic of Korea

Phone: +82-43-261-3664

Fax: +82-43-267-3150

Encl: Online submission

Response to reviewers

Reviewer 1

In this study, the authors focused on presenting iridoids as potential substances in the prevention and therapy of cancer. They mainly focused on the possibility of inhibiting the progression of various types of cancer, where references to in vitro studies were presented. Several important issues were discussed, incl. Anti-proliferative and apoptotic effects, Inhibitory effects on epithelial-mesenchymal transition, Inhibitory effects on cancer migration and invasion, Anti-antiangiogenic effects. The authors also highlighted many aspects that have not yet been addressed in teamt literature. Their exact list can be found in the conclusions. In my opinion, the work is interesting and contains key information on the mechanisms of anti-cancer action of natural substances. The article may be a good background for other researchers and a contribution to further research on not only iridoids but also other substances of natural origin. The work is well organized, however, in my opinion, the authors should pay more attention to the main subject of the work, which are iridoids. In my opinion, the introduction part should be divided into two sections. In the first part, the authors should devote more attention to the chemical nature of these compounds with the presentation of structural and derivative formulas and references to the species in which they occur. The second part should contain information about the biological activity of these compounds in connection not only with antitumor activity.

à Thanks for raising this valuable point to improve the quality of our manuscript. We agreed with the reviewer's suggestion to divide the introduction into two sections, and additionally described the contents that need to be supplemented in each part. In the first section of the introduction (Chemical nature of iridoid glycosides), the molecular and structural classification and biosynthesis of iridoids have been additionally described, and in the second section, the biological activities of iridoid glycosides have been further described as requested. These revisions have been marked in the colored text in the manuscript file (version R1).

Reviewer 2 Report

The paper offers an edge eys regarding the mechanisms involved in the antitumor effects of some iridoid glycosides present in herbal bioproducts (medicines or functional foods), by understanding the mechanisms of action of iridoid glycosides at each stage of cancer development. Based on data presented in literature published until the year 2021, the iridoid glycosides inhibit cancer proliferation by inducing cell cycle arrest or down-regulating the PI3K/Akt pathway 2 and the ERK1/2-cyclin D1 pathway, leading to programmed cell death or cytotoxicity. 
These classes of compounds (iridoid glycosides)  suppress the expression and activity of MMPs,  decrease the proteolytic activity for ECM,  with effect in decreasing cancer mobility and invasiveness More of that,  the antiangiogenic mechanism of iridoid glycosides is closely related to the transcriptional regulation of pro-angiogenic factors such as 286 VEGF and CD31. 
The paper is important for scientists which try to replace the synthetic molecules used in chemotherapy with other classes of compounds,  obtained from plants. From this point of view, the iridoid glycosides represent an important class of compounds,  and here, this paper can  can represent a starting point in elucidating the mechanisms of actions of iridoid glycosides. This is the reason for which I recommend  publishing this article, with minor revision, regarding the position of Table 1. 
Table 1 is more properly to be positioned before the Chapter entitled ''Conclusions''  ( respectively after row 274) with the following sentence: 
''Table 1 summarizes the effects and inhibitory mechanisms of iridoid glycosides at each stage of cancer development''. This sentence will be deleted from the Conclusions ( i.e. from the rows 287-288). 

Author Response

Aug 19, 2021

Editor-in Chief

Nutrients

Dear Editor-in Chief

The authors appreciate for valuable comments by the reviewers to complete the manuscript. The manuscript has been extensively revised based on the comments and a "Response to Reviewers" which addresses the individual comments of each reviewer has been attached.

Please find the revised manuscript (#nutrients-1329887.R1) entitled: “Potential roles of iridoid glycosides and their underlying mechanisms against diverse cancer growth and metastasis: Do they have an inhibitory effect on cancer progression?” by Kim CW and Choi KC, which we would like to re-submit as a review article for the Special Issue of the Nutrients.

 Special issue of “Antitumor Effects of Dietary Compounds”

Section of “Phytochemicals and Human Health”

I hope that everything will be found as required.

Yours Sincerely,

Kyung-Chul Choi, DVM, PhD

Laboratory of Veterinary Biochemistry and Immunology

College of Veterinary Medicine

Chungbuk National University

Cheongju, Chungbuk

361-763, Republic of Korea

Phone: +82-43-261-3664

Fax: +82-43-267-3150

Encl: Online submission

Response to reviewers

Reviewer 2

The paper offers an edge eys regarding the mechanisms involved in the antitumor effects of some iridoid glycosides present in herbal bioproducts (medicines or functional foods), by understanding the mechanisms of action of iridoid glycosides at each stage of cancer development. Based on data presented in literature published until the year 2021, the iridoid glycosides inhibit cancer proliferation by inducing cell cycle arrest or down-regulating the PI3K/Akt pathway 2 and the ERK1/2-cyclin D1 pathway, leading to programmed cell death or cytotoxicity.

These classes of compounds (iridoid glycosides) suppress the expression and activity of MMPs, decrease the proteolytic activity for ECM, with effect in decreasing cancer mobility and invasiveness More of that, the antiangiogenic mechanism of iridoid glycosides is closely related to the transcriptional regulation of pro-angiogenic factors such as 286 VEGF and CD31.

The paper is important for scientists which try to replace the synthetic molecules used in chemotherapy with other classes of compounds, obtained from plants. From this point of view, the iridoid glycosides represent an important class of compounds, and here, this paper can represent a starting point in elucidating the mechanisms of actions of iridoid glycosides. This is the reason for which I recommend publishing this article, with minor revision, regarding the position of Table 1.

Table 1 is more properly to be positioned before the Chapter entitled ''Conclusions'' ( respectively after row 274) with the following sentence:

''Table 1 summarizes the effects and inhibitory mechanisms of iridoid glycosides at each stage of cancer development''. This sentence will be deleted from the Conclusions ( i.e. from the rows 287-288).

à Thank you for raising this important point. Table 1 summarizes the anticancer mechanisms of iridoid glycosides described in the main body, thus we thought it would be better to place it at the end of main body. As per reviewer’s suggestion, Table 1 has been repositioned to the end of the main body from the ‘Conclusion and future perspectives’ chapter. In 'Conclusion and future perspectives' chapter, the sentence describing Table 1 (rows 287-288) has been deleted.

Reviewer 3 Report

There still have some issues need to be mention in this manuscript

  1. Author should describe more about the connection between iridoid glycosides and their related active ingredients
  2. The related mechanisms are varied, could author get the summarize of the related mechanism, does it have the specific target on the anti-cancer effect?
  3. Does all the ingredients are pure compound in these article?
  4. In the table, what’s the meaning of oleuropein (hydroxytyrosol)?
  5. The dosage of these compounds are same? Or it has the dose effect?
  6. The results need to have better organized and summarized. Since the title focused on the mechanism

Author Response

Aug 19, 2021

Editor-in Chief

Nutrients

Dear Editor-in Chief

The authors appreciate for valuable comments by the reviewers to complete the manuscript. The manuscript has been extensively revised based on the comments and a "Response to Reviewers" which addresses the individual comments of each reviewer has been attached.

Please find the revised manuscript (#nutrients-1329887.R1) entitled: “Potential roles of iridoid glycosides and their underlying mechanisms against diverse cancer growth and metastasis: Do they have an inhibitory effect on cancer progression?” by Kim CW and Choi KC, which we would like to re-submit as a review article for the Special Issue of the Nutrients.

 Special issue of “Antitumor Effects of Dietary Compounds”

Section of “Phytochemicals and Human Health”

I hope that everything will be found as required.

Yours Sincerely,

Kyung-Chul Choi, DVM, PhD

Laboratory of Veterinary Biochemistry and Immunology

College of Veterinary Medicine

Chungbuk National University

Cheongju, Chungbuk

361-763, Republic of Korea

Phone: +82-43-261-3664

Fax: +82-43-267-3150

Encl: Online submission

Response to reviewers

Reviewer 3

There still have some issues need to be mention in this manuscript

  1. Author should describe more about the connection between iridoid glycosides and their related active ingredients

à Thanks for your invaluable comments. This paper addresses the anticancer effects of compounds classified as iridoid glycosides, and they act as active ingredients themselves because they are single compounds extracted from medicinal plants or chemically synthesized.

  1. The related mechanisms are varied, could author get the summarize of the related mechanism, does it have the specific target on the anti-cancer effect?

à As you mentioned, the anticancer mechanisms of iridoid glycosides have been studied in many different ways, and the molecular biological signaling pathways are also diverse. Therefore, we tried to summarize these various mechanisms, and the inhibitory mechanisms of iridoid glycosides at each stage of cancer development can be briefly explained as follows. Iridoid glycosides inhibit cancer proliferation by inducing cell cycle arrest or down-regulating the PI3K/Akt pathway and the ERK1/2-cyclin D1 pathway, leading to programmed cell death or cytotoxicity. In addition, iridoid glycosides mainly suppress the expression and activity of MMPs, thereby lowering the proteolytic activity for ECM, resulting in significantly reduced cancer mobility and invasiveness. Moreover, the antiangiogenic mechanism of iridoid glycosides is closely related to the transcriptional regulation of pro-angiogenic factors such as VEGF and CD31. Taken together, these results indicate the therapeutic potential of iridoid glycosides to alleviate or prevent cascades of cancer development and metastasis. The ‘Conclusion and future perspectives’ chapter addresses these summarized mechanisms.

  1. Does all the ingredients are pure compound in these article?

à All ingredients covered in this article have been studied as pure compounds extracted from medicinal plants or chemically synthesized.

  1. In the table, what’s the meaning of oleuropein (hydroxytyrosol)?

à Hydroxytyrosol is one of the ester metabolites of oleuropein. It is an ingredient with anticancer potential, just like its parent compound, oleurpein. In the previous study, researchers tested the anticancer effects of oleuropein and hydroxytyrosol respectively, so the inhibitory mechanisms of oleuropein and hydroxytyrosol are summarized separately in Table 1. To avoid reader confusion, a brief description (annotation) of hydroxytyrosol has been added at the bottom of Table 1.

  1. The dosage of these compounds are same? Or it has the dose effect?

à Since the dose at which each iridoid glycoside exerts an anticancer effect is different, it can be considered that it has a dose effect.

  1. The results need to have better organized and summarized. Since the title focused on the mechanism.

à As you suggested, the results have been more organized and summarized with a focus on mechanisms.

Again, we appreciate all of your insightful comments. We have worked hard to respond to all comments as much as we could. Thank you for taking the time and contribution to help us improve this manuscript. Please let me know if there are any further corrections required. We look forward to hearing from you.

Round 2

Reviewer 1 Report

The authors only partially complied with my comments. In fact, they divided the introduction into two sections, but I hoped that more attention would be paid to the chemical nature of iridoids. In my opinion, the graphical representation of the structure of the ididoids is crucial for me to be able to accept the manuscript for publication

Author Response

Response to reviewers

Reviewer 1

The authors only partially complied with my comments. In fact, they divided the introduction into two sections, but I hoped that more attention would be paid to the chemical nature of iridoids. In my opinion, the graphical representation of the structure of the ididoids is crucial for me to be able to accept the manuscript for publication

à Thank you for your detailed comment. As per reviewer’s opinion, we have added the chemical structures of all iridoid glycosides mentioned in this review paper to Figure 1. In other words, a new Figure 1 has been created, and Figure Legend has been described on the last page of the manuscript (ver. R2).

 Reviewer 3

The author has summarized and reorganized the results, but the dosage of plant-derived ingredients may have a different aspect to the proliferative, EMT, or angiogenesis. My recommendation is author could add the dosage that the article used, respectively.

There still have some issues need to be mention in this manuscript

à Thank you for your valuable comments. The effective dosages of iridoid glycosides used in each study have been added to Table 1 (marked in purple), as you requested. This provides the reader with information about the specific concentrations exhibiting statistically significant anticancer effects of iridoid glycosides.

Reviewer 3 Report

The author has summarized and reorganized the results, but the dosage of plant-derived ingredients may have a different aspect to the proliferative, EMT, or angiogenesis. My recommendation is author could add the dosage that the article used, respectively.

Author Response

(The authors gave the same response as above.)
